# Endonasal Septoplasty Using a Septal Cartilaginous Batten Graft for Managing Caudal Septal Deviation

**DOI:** 10.3390/healthcare10091739

**Published:** 2022-09-11

**Authors:** Jessie Chao-Yun Chi, Shin-Da Lee, Chia-Yi Lee, Stanley Yung Liu, Hua Ting, Yih-Jeng Tsai

**Affiliations:** 1Institute of Medicine, Chung Shan Medical University, Taichung 40201, Taiwan; 2Department of Otorhinolaryngology, Head and Neck Surgery, Taichung Hospital, Ministry of Health and Welfare, Taichung 40343, Taiwan; 3Department of Physical Therapy, Graduate Institute of Rehabilitation Science, China Medical University, Taichung 40402, Taiwan; 4Nobel Eye Institute, Taipei 100008, Taiwan; 5Department of Ophthalmology, Jen-Ai Hospital Dali Branch, Taichung 41265, Taiwan; 6Department of Otolaryngology, Stanford University School of Medicine, Stanford, CA 94305, USA; 7Department of Physical Medicine and Rehabilitation, Chung Shan Medical University Hospital, Taichung 402367, Taiwan; 8Sleep Medicine Center, Chung Shan Medical University Hospital, Taichung 402367, Taiwan; 9School of Medicine, Fu-Jen Catholic University, New Taipei City 24205, Taiwan; 10Department of Otolaryngology Head and Neck Surgery, Shin Kong Wu Ho-Su Memorial Hospital, Taipei 111045, Taiwan

**Keywords:** caudal septal deviation, endonasal septoplasty, functional rhinoplasty, septal cartilaginous batten graft, autologous graft

## Abstract

Caudal nasal septal deviation is an important condition altering nasal obstruction and cosmetic appearance and many surgical techniques have been published on how to correct caudal septal deviation, as successful management of caudal septal deviation is challenging. The goal of our study was to explore the effect of endonasal septoplasty using a septal cartilaginous batten graft for managing caudal septal deviation. We tested 26 participants with caudal septal deviation who received endonasal septoplasty using a septal cartilaginous batten graft from 1 April 2019 to 29 June 2022, and followed up for at least 6 months. Nasal Obstruction Symptom Evaluation (NOSE) Scale and visual analog scale (VAS) were recorded at baseline, 1 month, and 6 months after surgery. Valid samples were analyzed by repeated measures ANOVA and paired sample *t*-test. Average participant age was 36.15 ± 11.02 years old. The preoperative, 1-month postoperative, and 6-month postoperative NOSE scale decreased significantly (75.38 ± 15.62, 13.85 ± 7.79, and 14.04 ± 9.90; *p* < 0.001), while preoperative, 1-month postoperative, and 6-month postoperative VAS (convex/concave side) also improved (7.50 ± 0.81/3.38 ± 0.94, 2.27 ± 0.53/1.54 ± 0.58, and 2.31 ± 0.55/1.58 ± 0.58; *p* < 0.001). Our results showed that endonasal septoplasty using a septal cartilaginous batten graft had good surgical outcomes without an open scar or severe complications.

## 1. Introduction

Nasal obstruction is a very common and annoying symptom often caused by physical obstruction of nasal passages and related physiological problems, such as rhinitis and rhinosinusitis [1]. Treatment includes medication, such as antihistamine, decongestants, anticholinergics, intranasal and systemic steroids [1,2], and nasal surgery to overcome the anatomic obstruction [3]. From an anatomic perspective, a deviated nasal septum is one of the major causes [3], especially caudal nasal septal deviation [3,4]. Nowadays, numerous surgical techniques are used to correct caudal septal deviation, as successful management of caudal septal deviation remains challenging [3,5]. It is difficult to maintain the function of the L strut of the septal cartilage and overcome the intrinsic cartilage memory at the same time [3]. Septoplasty for managing caudal septal deviation can involve a cartilage batten graft with or without multiple incisions and wedge resection of the caudal septal cartilage [4]. Septoplasty by cartilage batten graft is a simple surgical technique, but sometimes can require correction and lead to persistent nasal obstruction [3], as well as a narrower contralateral nasal cavity due to a thick batten graft [3]. In some cases, extracorporeal septoplasty via open rhinoplasty is required [6]. However, postoperative problems after open rhinoplasty was twice as frequent as after an endonasal approach in patients with revision surgery [7]. Compared with an endonasal approach, patients who received open rhinoplasty showed alar collapse (50%), over rotation of the tip (39%), a wide columella base (36%), collapsed cartilaginous dorsum (31%), visible columella scars (25%), and a columella transplant with discomfort (19%) [7]. Many studies have also described the use of bony batten grafts, harvested from the perpendicular plate of the ethmoid bone or the vomer bone, to correct the deviated caudal septum [8,9,10,11,12]. However, drilling bone grafts may lead to breakage [9]. Surgeons have frequently attempted to avoid under- or overcorrection of caudal cartilage deviation and alleviated nasal obstruction without subsequent nasal deformity [13].

Based on these studies, we proposed to derive an effective surgical technique to correct caudal septal deviation with better outcomes and fewer complications. In our surgical technique, no bony cut, drilling, scoring, difficult suturing, or open rhinoplastic approach was required. We also used a nasal septal cartilaginous batten graft to perform the endonasal septoplasty, eliminating the need for other donor sites, such as ear cartilage, costal cartilage, or the ethmoid and vomer bone.

## 2. Materials and Methods

### 2.1. Study Design

This retrospective study was conducted at a single hospital from 1 April 2019 to 29 June 2022. The study was approved by the Institutional Review Board of the Institutional Review Board of the Tsao-tun Psychiatric Center, Taichung, Taiwan (protocol code: No. 110011; Date of approval: 11 April 2021). The reporting guideline that was followed in our study and the informed consent statement was waived due to the retrospective study.

### 2.2. Study Population

The inclusion criteria were (1) adult age (>20 years old), (2) severe nasal obstruction that failed to respond to medical treatment for more than three months, and (3) significant caudal nasal septal deviation confirmed by nasal endoscopy. The exclusion criteria were (1) patients with uncontrolled coagulation disorders or other severe underlying diseases, (2) a history of allergic reactions to local anesthetics, (3) failure to follow up 6 months after surgery, and (4) insufficient septal cartilage for a septal batten graft. In our study, 26 participants who had a caudal septal deviation and received endonasal septoplasty using an autologous septal cartilaginous batten graft and bilateral inferior turbinoplasty were recruited from our Otorhinolaryngology Head and Neck Surgery clinics and followed for at least 6 months.

### 2.3. Surgical Techniques

In this article, we named the “endonasal” because it is mentioned in the textbook [14,15]. If the columella is not incised, it is considered an endonasal; and if it is incised, it is considered an open approach. In our surgical technique, no bony cut, drilling, scoring, difficult suturing, or open rhinoplastic approach was required. We also used an endoscope when we did middle, posterior segments of the nasal septum, and inferior turbinoplasty.

The caudal nasal septum, which was deviated to the left side (Figure 1A1,A2), was exposed using a traditional hemitransfixion incision (Figure 1B) and the bilateral mucoperichondrial flaps were carefully elevated (Figure 1C) to free the quadrangular cartilage (entire cartilage in the first surgery and residue cartilage in the revised surgery), anterior nasal spine, bony-cartilage junction between the septal cartilage and bony septum, perpendicular plate of the ethmoid bone, and vomer bone. A hemitransfixion incision was performed from the left side of the nasal cavity irrespective of left or right side deviation of the caudal nasal septum. The middle and posterior portions of the nasal septal deviated cartilage were harvested and reshaped. A 1.5 cm section of the L strut of the dorsal cartilaginous septum was left in place, while only a 1.0 cm section of the L strut of the caudal cartilaginous septum was preserved, to avoid septal cartilage intrinsic memory after functional rhinoplasty. The height of the cartilage batten graft covered the entire L-shaped caudal cartilage, and its width was a minimum 2.0 cm in the primary septoplasty and a minimum 1.5 cm in the revised septoplasty due to the lack of septal cartilage in revised septoplasty patients.

Autologous septal cartilaginous batten grafts were reshaped and reconstructed in the concave side of the caudal nasal septum. As seen in Figure 1D1, the batten graft was reconstructed in the right (concave) side of the nose, and Figure 1D2 shows the batten graft as supported in the left (concave) side of the nose. The L strut cartilage was preserved for at least 1.0 cm from the caudal end and the connection between the septal cartilage, and the anterior nasal spine was also maintained. There were no incisions or wedge resections of the caudal nasal septum. The L-shaped caudal end and the cartilage batten graft were connected via through-and-through sutures with three stitches using 5-0 Prolene sutures in the concave side (Figure 1D2). Finally, the mucoperichondrial flaps were repositioned and 5-0 Vicryl simple sutures were used to close the incision wound (Figure 1E). Bilateral inferior turbinoplasty was also performed in all 26 studied cases. Absorbable nasal packing (nasopore) was packed bilaterally. Figure 2A1 shows the caudal septal deviation to the left side, and Figure 2B1 shows the right side. We reshaped and reconstructed the cartilaginous batten graft for the concave side (Figure 2A2,B2). We applied three stitches using through-and-through sutures to fix the batten graft and 5-0 Prolene sutures for the caudal end of the L strut. The nodes were fixed in the concave side (Figure 2A3,B3). When we performed surgery on the middle and posterior segments of the nasal septum, we used an endoscope. We also used an endoscope when we did inferior turbinoplasty.

### 2.4. Protocol

Information related to anthropometric measurements, nasal anatomy, current medications, general health data, surgery history, and clinical symptoms, as well as otorhinolaryngological examinations and services, were obtained on the occasion of the clinical visit, the time points for nasoscopy, and chart recording. All surgical procedures were performed by one otorhinolaryngologist (the first author of this article). For surgical outcome assessment, the nasal obstruction symptom evaluation scale (NOSE scale) and visual analog scale (VAS scale) were recorded.

To evaluate outcomes preoperative, 1-month postoperative, and 6-month postoperative, we used the NOSE scale based on chart records and VAS using flexible nasopharyngoscopic graphs. The definition of VAS of the convex side measured distance from the largest protruding part of the caudal nasal septal region (convex-most part) to the midline (Figure 3a) divided by the distance of the unilateral nasal floor (Figure 3b). The VAS of the concave side was also measured as the largest protruding part to the midline (Figure 3a’), which was divided by the distance of the unilateral nasal floor (Figure 3b’). All data were multiplied 10-fold for calculation and analysis (Convex a/b × 10; Concave a’/b’ × 10).

### 2.5. Statistical Analyses

Statistical analysis was performed using SPSS 25.0 (IBM, Armonk, NY, USA). A significance threshold of *p* < 0.05 was used. Repeated measures ANOVA and paired sample t-tests were used for data analysis (NOSE scale and VAS).

## 3. Results

Among the 26 patients (23 male, 3 female; in the following results, M/F), the average age was 36.15 ± 11.02 years. A total of 23 participants presented with primary functional rhinoplasty (88.46%) and the remaining three (11.54%) presented with revision surgery. The preoperative, 1-month postoperative, and 6-month postoperative NOSE scale values were 75.38 ± 15.62, 13.85 ± 7.79, and 14.04 ± 9.90 (*p* < 0.001). The reduction rate of NOSE scale was 81.63% for the post-operative first month and 81.37% for the post-operative sixth month. The preoperative, 1-month postoperative, and 6-month postoperative VAS (convex/concave side) were 7.50 ± 0.81/3.38 ± 0.94, 2.27 ± 0.53/1.54 ± 0.58, and 2.31 ± 0.55/1.58 ± 0.58 (*p* < 0.001). The VAS reduction rate was 69.73%/54.44% (convex/concave side) for the post-operative first month and 69.20%/53.25% (convex/concave side) for the post-operative sixth month. Concerning confounding factors, among 26 participants, no patient required long-term use of nasal sprays after the surgery. Regarding underlying diseases, two participants had asthma, one participant smoked, one participant had cleft palate s/*p* operation, and five participants had obstructive sleep apnea.

Preoperative status showed significant improvements in both NOSE scale and VAS after endonasal functional rhinoplasty using an autologous septal cartilaginous batten graft (Table 1 and Figure 4) for both postoperative time points. However, when comparing postoperative 1 month and 6 months, there was no significant difference in neither NOSE scale (*t*-test; *p* = 0.862) or VAS (*p* = 0.327 for the convex side; *p* = 0.664 for the concave side). During our follow-up period, no patients experienced severe complications, such as saddle nose deformity, nasal tip collapse, or septal hematoma, and there were no other complications such as narrowing concave side due to batten graft, septal abscess, infection, septal perforation, severe bleeding, graft extrusion, nasal cavity granulation tissue formation, hyposmia, or nasal tip numbness [3,16].

## 4. Discussion

Many surgical techniques have been published on how to correct caudal septal deviation, as successful management of caudal septal deviation is challenging. The goal of our study was to modify caudal septal deviation correction surgery and reach a reliable surgical outcome using a simple and effective surgical technique. We found that endonasal septoplasty using a cartilaginous septal batten graft was an effective surgical technique without severe complications. The reduction rate of the NOSE scale was 81.63% for the post-operative first month and 81.37% for the post-operative sixth month, while the VAS reduction rate was 69.73/54.44% (convex/concave side) for the post-operative first month and 69.20/53.25% (convex/concave side) for the post-operative sixth month.

Reviewing previous studies, seven articles published in English use NOSE scale statistics [8,9,17,18,19,20,21], showing a minimal postoperative NOSE improvement rate of 55.50% [17] and a maximum of 88.89% [9]. Among these studies, two articles reported improvements by more than 80% [9,19]. In 2016, Yunus Karadavut et al. [19] used caudal septal extension graft application in endonasal septoplasty and recorded a NOSE scale improvement from 80 to 15, with reduction rate of 81.25%. The study used ear or costal cartilage as a septal extension graft for all patients (*n* = 20). In 2018, Y-C Lee et al. [9] used bone batten graft and recorded a NOSE scale reduction from 72 to 8 and the reduction rate was 88.89% (*n* = 22). They harvested the hard bone of the nasal septum as the batten graft. However, in our study, we used a nasal septal cartilaginous batten graft to perform the endonasal septoplasty, eliminating the need for other donor sites, such as ear cartilage, costal cartilage, or the ethmoid and vomer bone. We reshaped the deviated nasal septal cartilage and reconstructed it as the autologous cartilaginous batten graft using an endonasal approach without an open scar. We achieved a NOSE scale reduction rate of 81.63% at 1 month post operation and 81.37% at 6 months post operation. Our results show that endonasal septoplasty using a septal cartilaginous batten graft is a simple surgical technique for managing caudal septal deviation with good NOSE scale outcomes compared with previous studies.

Two previous studies considered the early and late postoperative NOSE scales. In 2015, Josh Surowitz et al. [21] used anterior septal reconstruction to treat severe caudal septal deviation in 77 patients. The reduction rates of NOSE scale were 69.06% (post-op 1.4-month) and 76.83% (post-op 7.5-month) when using extracorporeal and open rhinoplasty. The authors noted this improvement in NOSE scale over time. However, due to some of the 77 participants missing regular follow up, only 75 participants in the early postoperative NOSE scale (post-op 1.4-month) and 41 participants in the late postoperative NOSE scale (post-op 7.5-month) were examined, which may have introduced bias. In another study in 2017, Do-Youn Kim et al. [8] published surgical outcomes of bony batten grafting to correct caudal septal deviation in septoplasty. Of 141 participants who attended follow-up, the reduction rates of NOSE scale were 62.41% (post-op 2-month) and 59.29% (post-op 6-month). The authors noted the NOSE scale could be slightly elevated over time, an effect which we also saw in our study. We suggest that this could be due to cartilage intrinsic memory.

Concerning objective nasal cavity space survey, some studies objectively measured nasal cavity volume and cross section area by acoustic rhinometer [19], facial angle calculation [19], and nasosinus computed tomography [9]. In our study, we used VAS scores to monitor objective anatomic changes by flexible nasopharyngoscope, providing a simple, effective, and relatively cheap method for evaluating nasal space change. We noted that the reduction rate of VAS was 69.73/54.44% (convex/concave side) for the post-operative first month and 69.20/53.25% (convex/concave side) for the post-operative sixth month. This surgical technique allowed us to improve both convex and concave sides simultaneously.

Caudal septal deviation cannot be corrected delicately using traditional septomeatoplasty, and many surgical techniques have been published to address this issue. In 2020, Béatrice Voizard et al. published a systematic review on caudal septoplasty in North American case studies [5]. They found that the most common techniques were the swing door technique (69.5%), extracorporeal septoplasty (46.7%), cartilage scoring (45.3%), and splinting with bone (25.4%). The most popular swing door technique uses cartilage reshaping and repositioning techniques for caudal septal dislocations, and applies an anatomic re-orientation between the septum and nasal spine for caudal septal deviations and subluxations [17]. However, re-orientating between the septum and nasal spine introduces the risk of dislocation and subsequent surgical failure [22]. Extracorporeal septoplasty has been suggested as an effective approach for both functional and cosmetic treatment of moderate to severe deformities of the caudal and dorsal septum [6,23]. A review on extracorporeal septoplasty showed that reported complication rates were low, but tip deprojection and rotation were observed [23]. Concerning bony batten graft, it may be much easier than extracorporeal septoplasty, but it still requires time to harvest and reshape the bony graft and drilling bone grafts may lead to breakage. [9,10]. For cartilage scoring, most techniques were combined with the batten graft [17,24]. Compared to traditional Cottle’s operation, although it could improve nasal valve stenosis, unfortunately, it could not correct the caudal septal deviation and anterior nasal spine delicately [3,25,26]. No matter nasal valve correction or caudal septal deviation correction, both of them could improve nasal space. Determining the reasons for nasal obstruction could be the most important thing. The patients who had caudal septal deviation may choose our surgical technique because it can correct caudal septal cartilage and anterior nasal spine delicately and successfully instead of traditional Cottle’s operation. Therefore, preoperative diagnosis is very important to accurately determine reasons of nasal obstruction and select appropriate surgical techniques for better postoperative outcomes.

In our surgical technique, no bony cut, drilling, scoring, difficult suturing, or open rhinoplastic approach was required, as the cartilaginous batten graft was easy to harvest and reshape from the middle and posterior parts of the nasal septal cartilage. The two key points of our surgical technique are (1) the height of the cartilage batten graft covered the entire L-shaped caudal cartilage, and its width was a minimum 2.0 cm in the primary septoplasty and a minimum of 1.5 cm in the revised septoplasty; (2) only a 1.0 cm section of the L strut of the caudal cartilaginous septum was preserved to avoid septal cartilage intrinsic memory after functional rhinoplasty. We performed endonasal functional rhinoplasty using a cartilaginous septal batten, resulting in good outcomes on the NOSE and VAS scales 1 month after surgery, which even persisted for at least 6 months. We found no potential rhinoplasty complications as we mentioned before. Although our article is similar to the number and NOSE score of patients in other published papers [9,19], a limitation of our study that should be noted is its retrospective character with a short follow up period, the small sample size, and high NOSE score, which may have led to potential bias. Because the sample size is small, analysis of subgroups makes little sense. We will continue to study and expand the number of cases, attempt to include a control group and low NOSE score, and perform an analysis of subgroups in the future.

## 5. Conclusions

In summary, we found that endonasal septoplasty using an autologous septal cartilaginous batten graft to treat caudal nasal septal deviation can be a useful technique that may be performed with relative ease and simplicity. Our results showed that endonasal septoplasty using a septal cartilaginous batten graft had good surgical outcomes without an open scar or severe complications. This technique can improve the NOSE scale subjectively and VAS objectively for at least 6 months. A longer-term follow-up period can be explored in future studies.

## Figures and Tables

**Figure 1 healthcare-10-01739-f001:**
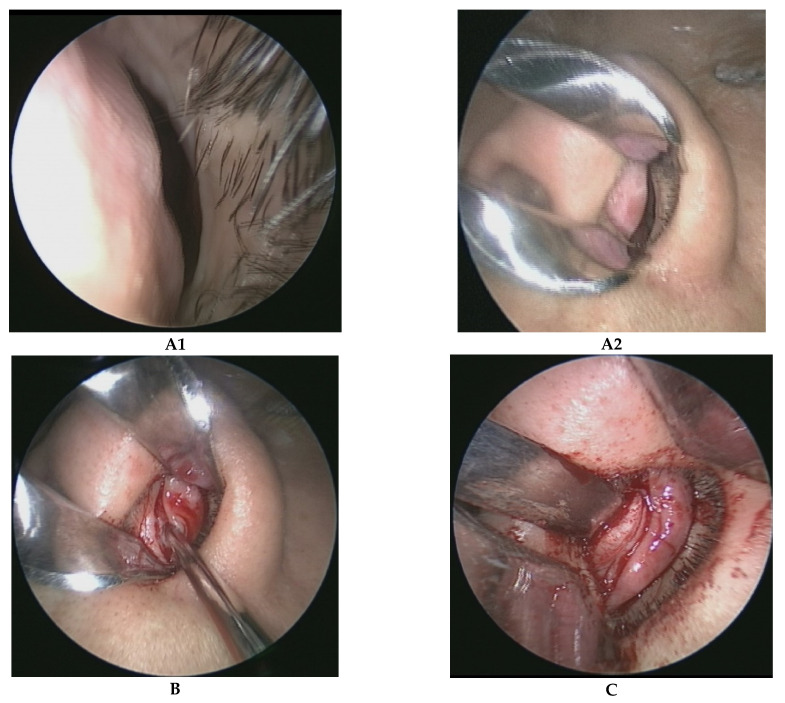
A step-by-step illustration of the surgical techniques used.

**Figure 2 healthcare-10-01739-f002:**
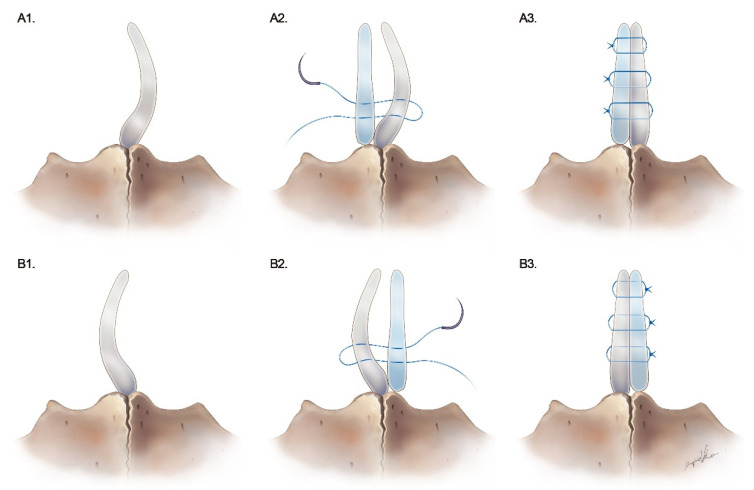
Schematic illustration of a cartilaginous batten graft with suture techniques.

**Figure 3 healthcare-10-01739-f003:**
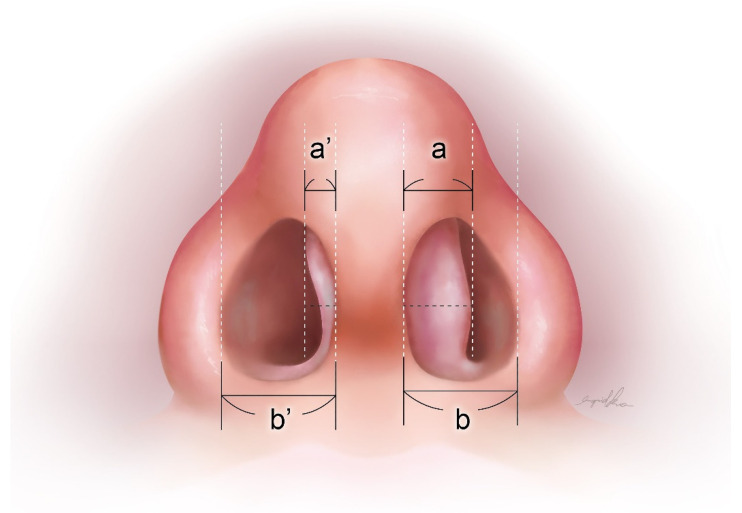
Illustration of calculation of visual analog scale (VAS).

**Figure 4 healthcare-10-01739-f004:**
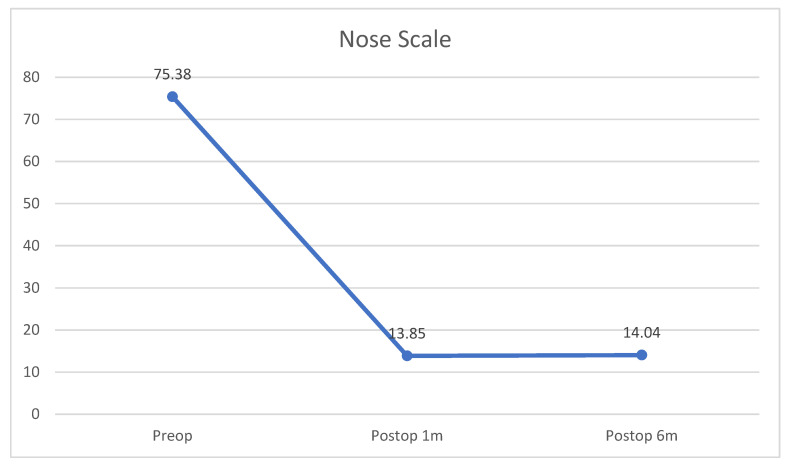
Nasal obstruction symptom evaluation (NOSE) scale and visual analog scale (VAS) improvement after performing endonasal functional rhinoplasty using an autologous septal cartilaginous batten graft. Abbreviations: NOSE, nasal obstruction symptom evaluation; VAS, visual analog scale; 1m, 1 month; 6m, 6 months.

**Table 1 healthcare-10-01739-t001:** Symptoms and nasal cavity space improvement 1 month and 6 months after performing endonasal functional rhinoplasty using an autologous septal cartilaginous batten graft.

Outcome Index	Preoperative	Postoperative (1 Month)	Postoperative (6 Months)	P1	P2
NOSE Scale [Mean ± SD]	75.38 ± 15.62	13.85 ± 7.79	14.04 ± 9.90	<0.001 *	*p* = 0.862
VAS (Convex side) [Mean ± SD]	7.50 ± 0.81	2.27 ± 0.53	2.31 ± 0.55	<0.001 *	*p* = 0.327
VAS (Concave side) [Mean ± SD]	3.38 ± 0.94	1.54 ± 0.58	1.58 ± 0.58	<0.001 *	*p* = 0.664

Abbreviations: NOSE, nasal obstruction symptom evaluation; VAS, visual analog scale; SD, standard deviation. * Significant of the outcome parameters among different time points. P1, repeated ANOVA in these three groups; P2, paired T test in the postoperative (one month) and postoperative (6 months).

## Data Availability

The data presented in this study are available on request from the corresponding author. The data are not publicly available because we will perform further studies in the future.

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
