# Peer review of "Endonasal Septoplasty Using a Septal Cartilaginous Batten Graft for Managing Caudal Septal Deviation"

_healthcare, 2022, doi:10.3390/healthcare10091739_

Round 1

Reviewer 1 Report

My major comment is that, as in many papers where another technique is presented, some caveats need to be made when presenting the study. 1) sample size is small; 2) a selection bias may have occurred since only patients with severe nasal obstruction were selected (high NOSE score meaning that the benefit would have been found whatever the septoplasty technique used); 3) no comparison were found in the discussion between this technique versus others: e.g., what are its advantages/disadvantages compared to traditional Cottle's operation? who is the ideal candidate for this technique?; 4) confounding factors should be described at least and accounted for in subanalyses: use of postop intranasal steroids? presence of respiratory allergy? smoking status?

Finally,  I disagree on calling this technique "endonasal functional rhinoplasty": after all, it's just another classic open non-endoscopic septoplasty technique

Author Response

Dear Reviewer,

  We are pleased to resubmit the revised article to HealthCare –Special Issue: "2nd Edition of Advances in Otolaryngology from Diagnosis to Treatment" for consideration for publication. We have revised the manuscript according to the suggestions from the editor and the reviewers and have responded to the reviewer comments. The changes we made are highlighted in the revised version using tracked changes, and the revised manuscript conforms to the journal style. We hope that the responses address the requests and suggestions and that our manuscript can be accepted by HealthCare –Special Issue: "2nd Edition of Advances in Otolaryngology from Diagnosis to Treatment".

Thank you for your professional advice and wish you have a nice day~

Regards,

Jessie Chao-Yun Chi; Shin-Da Lee; Chia-Yi Lee; Stanley Yung Liu; Hua Ting*; Yih-Jeng Tsai* (*Corresponding author)

Reviewer

  • Sample size is small

<Response>

Thank you for the valuable suggestion. We have already added this part in the limitation of the article.

<Revision>

Pg. 10; Lns 4-8. Although our article is similar to the number and NOSE score of patients in other published papers [9,19], a limitation of our study that should be noted is its retrospective character with a short follow up period, the small sample size, and high NOSE score which may have led to potential bias.

  • A selection bias may have occurred since only patients with severe nasal obstruction were selected (high NOSE score meaning that the benefit would have been found whatever the septoplasty technique used)

<Response>

Thank you for the valuable suggestion. We have already added this part in the limitation of the article.

<Revision>

Pg. 10; Lns 4-8. Although our article is similar to the number and NOSE score of patients in other published papers [9,19], a limitation of our study that should be noted is its retrospective character with a short follow up period, the small sample size, and high NOSE score which may have led to potential bias.

  • No comparison were found in the discussion between this techniques versus others: e.g., what are its advantages/disadvantages compared to traditional Cottle's operation? who is the ideal candidate for this technique?

<Response>

Thank you for the valuable suggestion. We have already compared the other published articles in surgical techniques on the correction of the caudal nasal septum [3,6-12], but we didn't compare the difference between traditional Cottle's operation and our surgical technique. Therefore, we have included this part in the discussion of the article.

<Revision>

Pg. 9; Lns 40-47. Compared to traditional Cottle's operation, although it could improve nasal valve stenosis, unfortunately, it couldn’t correct the caudal septal deviation and anterior nasal spine delicately [3,27,28]. No matter nasal valve correction or caudal septal deviation correction, both of them could improve nasal space. Determining the reasons for nasal obstruction could be the most important thing. The patients who had caudal septal deviation may choose our surgical technique because it can correct caudal septal cartilage and anterior nasal spine delicately and successfully instead of traditional Cottle’s operation.

  • Confounding factors should be described at least and accounted for in subanalyses: use of postop intranasal steroids? presence of respiratory allergy? smoking status?

<Response>

Thank you for the valuable suggestion. We have included this part in the “result” section and “limitation” section.

<Revision>

Pg. 6; Lns 16-18. Concerning confounding factors, among 26 participants, those who did not use nasal sprays long-term postoperatively. Regarding underlying diseases, 2 participants had asthma, 1 participant smoked, 1 participant had cleft palate s/p operation, and 5 participants had obstructive sleep apnea. 

Pg. 10; Lns 8-9. Because the sample size is small, analysis of subgroups makes little sense. We will continue to study and expand the number of cases received in the future.”

  • Finally, I disagree on calling this technique "endonasal functional rhinoplasty": after all, it's just another classic open non-endoscopic septoplasty technique

<Response>

Thank you for the valuable suggestion. First of all, I want to apologize for not being clear in the article. When we performed surgery on the middle and posterior segments of the nasal septum, we used an endoscope. We also used an endoscope when we did inferior turbinoplasty. Therefore, we have included this part in the “surgical techniques” of the article.

We named the “endonasal” because it is mentioned in the textbook [14, 15]. If the columella is not incised, it is considered an endonasal; and if it is incised, it is considered an open approach. Therefore, we added this paragraph to the “surgical techniques” to make the article much clear.

<Revision>

Pg. 3; Lns 24-26. When we performed surgery on the middle and posterior segments of the nasal septum, we used an endoscope. We also used an endoscope when we did inferior turbinoplasty.

Pg. 2; Lns 40-42. In this article, we named the “endonasal” because it is mentioned in the textbook [14,15]. If the columella is not incised, it is considered an endonasal; and if it is incised, it is considered an open approach.

Reviewer 2 Report

I would like to thank the authors for the opportunity of having such a good piece of research for me to read and evaluate.

The authors are trying to evaluate the effect of endonasal septoplasty using cartilaginous batten graft for managing caudal septal deviation. Between 2019 and 2022 26 patients have been enrolled. NOSE and VAS scales have been applied to evaluate the effect of the surgical procedure. The results have been statistically evaluated at preoperative, 1 month and 6 months postop. The results show good improvement after the surgical septoplasty.

The research protocol is outstanding and checks all the boxes regarding proper procedures. The authors very good explain the surgical technique using illustrations and clinical images. Objectively using a scale such as NOSE and VAS helps the reader understand and critically evaluate and understand the results.

The discussion paragraph is very good written, with comparison with similar studies which use the NOSA and VAS scales. 

We strongly recommend the publication of this original research article in the current form.

Author Response

Dear Reviewer,

  We are pleased to resubmit the revised article to HealthCare –Special Issue: "2nd Edition of Advances in Otolaryngology from Diagnosis to Treatment" for consideration for publication. We have revised the manuscript according to the suggestions from the editor and the reviewers and have responded to the reviewer comments. The changes we made are highlighted in the revised version using tracked changes, and the revised manuscript conforms to the journal style. We hope that the responses address the requests and suggestions and that our manuscript can be accepted by HealthCare –Special Issue: "2nd Edition of Advances in Otolaryngology from Diagnosis to Treatment".

Thank you for your professional advice and wish you have a nice day~

Regards,

Jessie Chao-Yun Chi; Shin-Da Lee; Chia-Yi Lee; Stanley Yung Liu; Hua Ting*; Yih-Jeng Tsai* (*Corresponding author)

Reviewer

“The research protocol is outstanding and checks all the boxes regarding proper procedures. The authors very good explain the surgical technique using illustrations and clinical images. Objectively using a scale such as NOSE and VAS helps the reader understand and critically evaluate and understand the results.

The discussion paragraph is very good written, with comparison with similar studies which use the NOSA and VAS scales.  

We strongly recommend the publication of this original research article in the current form.”

<Response>

Thank you very much for your affirmation and it is our honor to ask for help from the expert. Our team will continue to work hard and look forward to communicating with you in the future!

Reviewer 3 Report

Successful management of caudal septal deviation is challenging, however,there is some innovation and novelty in this study. At the same time, I also pay attention to two issues: the first is that the caudal part of the nasal septum will be widened after a septal cartilaginous batten graft, will the nasal caudal part be restored or close to the original width in the future? The second is whether the caudal cartilage will be necrotic, leading to nasal tip collapse?

Author Response

Dear Reviewer,

  We are pleased to resubmit the revised article to HealthCare –Special Issue: "2nd Edition of Advances in Otolaryngology from Diagnosis to Treatment" for consideration for publication. We have revised the manuscript according to the suggestions from the editor and the reviewers and have responded to the reviewer comments. The changes we made are highlighted in the revised version using tracked changes, and the revised manuscript conforms to the journal style. We hope that the responses address the requests and suggestions and that our manuscript can be accepted by HealthCare –Special Issue: "2nd Edition of Advances in Otolaryngology from Diagnosis to Treatment".

Thank you for your professional advice and wish you have a nice day~

Regards,

Jessie Chao-Yun Chi; Shin-Da Lee; Chia-Yi Lee; Stanley Yung Liu; Hua Ting*; Yih-Jeng Tsai* (*Corresponding author)

Reviewer

  1. The caudal part of the nasal septum will be widened after a septal cartilaginous batten graft, will the nasal caudal part be restored or close to the original width in the future?

<Response>

Thank you for the valuable suggestion. Septoplasty by cartilage batten graft sometimes may lead to a narrower contralateral nasal cavity due to a thick batten graft [3]. Therefore, to objectively measure the impact of this part, in the measurement objective space, we not only calculate VAS convex but also VAS concave in the “protocol” section. We specially add this part-“No narrowing concave side due to batten graft” to the “result” section.

<Revision>

Pg. 6; Lns 24-26. Pg. 7; Lns 1-2. During our follow-up period, no patients experienced severe complications, such as saddle nose deformity, nasal tip collapse, or septal hematoma, and there were no other complications such as narrowing concave side due to batten graft, septal abscess, infection, septal perforation, severe bleeding, graft extrusion, nasal cavity granulation tissue formation, hyposmia, or nasal tip numbness [3,16].

  1. Whether the caudal cartilage will be necrotic, leading to nasal tip collapse?

<Response>

Thank you for the valuable suggestion. The caudal cartilage will be necrotic, leading to nasal tip collapse [3, 7, 13]. Compared with an endonasal approach, patients who received open rhinoplasty showed alar collapse (50%), over rotation of the tip (39%), a wide columella base (36%), collapsed cartilaginous dorsum (31%), visible columella scars (25%), and a columella transplant with discomfort (19%) [7]. In our case, no such sequelae occurred and we specially add this part-“No nasal tip collapse” to the “result” section.

<Revision>

Pg. 6; Lns 24-26. Pg. 7; Lns 1-2. During our follow-up period, no patients experienced severe complications, such as saddle nose deformity, nasal tip collapse, or septal hematoma, and there were no other complications such as narrowing concave side due to batten graft, septal abscess, infection, septal perforation, severe bleeding, graft extrusion, nasal cavity granulation tissue formation, hyposmia, or nasal tip numbness [3,16].

Round 2

Reviewer 1 Report

Despite the revisions, many criticalities are still unresolved in this paper.

Author Response

Dear Reviewer,

  We are pleased to resubmit the revised article to HealthCare –Special Issue: "2nd Edition of Advances in Otolaryngology from Diagnosis to Treatment" for consideration for publication. We have revised the manuscript according to the suggestions from the editor and the reviewers and have responded to the reviewer comments. The changes we made are highlighted in the revised version using tracked changes, and the revised manuscript conforms to the journal style. We hope that the responses address the requests and suggestions and that our manuscript can be accepted by HealthCare –Special Issue: "2nd Edition of Advances in Otolaryngology from Diagnosis to Treatment".

Thank you for your professional advice and wish you have a nice day~

Regards,

Jessie Chao-Yun Chi; Shin-Da Lee; Chia-Yi Lee; Stanley Yung Liu; Hua Ting*; Yih-Jeng Tsai* (*Corresponding author)

Reviewer

  1. English language and style

(x) Extensive editing of English language and style required
<Response>

Thank you for your variable suggestion. Our article has been revised in English by “Uni-edit” and presented in the “Acknowledgments” section- “The authors wish to thank Uni-edit for editing and proofreading this manuscript”.

<Basic Data>

University English Editing and Translation Service (Uni-edit)

[email protected]

Skype: uni-edit-huang

direct cell 0975 830 877

www.uni-edit.net

Subject: Uni-edit quotation for C2605002TAIC (L3) 

If it is necessary, we can use the English revision company recommended by HealthCare.

  1. Despite the revisions, many criticalities are still unresolved in this paper.

<Response>

Thank you for the valuable suggestion. We have already explain further in the limitation of the article. At the same time, we also add some explanations and annotations in other paragraphs, hoping to meet the requirements as much as possible.

<Revision>

Pg. 10; Lns 4-10. Although our article is similar to the number and NOSE score of patients in other published papers [9,19], a limitation of our study that should be noted is its retrospective character with a short follow up period, the small sample size, and high NOSE score which may have led to potential bias. Because the sample size is small, analysis of subgroups makes little sense. We will continue to study and expand the number of cases, attempt to include a control group and low NOSE score, and analysis of subgroups in the future.

Pg. 2; Lns 19-24. Based on these studies, we proposed to derive an effective surgical technique to correct caudal septal deviation with better outcomes and fewer complications. In our surgical technique, no bony cut, drilling, scoring, difficult suturing, or open rhinoplastic approach was required. We also used a nasal septal cartilaginous batten graft to perform the endonasal septoplasty, eliminating the need for other donor sites, such as ear cartilage, costal cartilage, or the ethmoid and vomer bone.

Pg. 2; Lns 44-48. In this article, we named the “endonasal” because it is mentioned in the textbook [14,15]. If the columella is not incised, it is considered an endonasal; and if it is incised, it is considered an open approach. In our surgical technique, no bony cut, drilling, scoring, difficult suturing, or open rhinoplastic approach was required. We also used an endoscope when we did middle, posterior segments of the nasal septum, and inferior turbinoplasty.

Pg. 9; Lns 40-50. Compared to traditional Cottle's operation, although it could improve nasal valve stenosis, unfortunately, it couldn’t correct the caudal septal deviation and anterior nasal spine delicately [3,27,28]. No matter nasal valve correction or caudal septal deviation correction, both of them could improve nasal space. Determining the reasons for nasal obstruction could be the most important thing. The patients who had caudal septal deviation may choose our surgical technique because it can correct caudal septal cartilage and anterior nasal spine delicately and successfully instead of traditional Cottle’s operation. Therefore, preoperative diagnosis is very important to accurately determine reasons of nasal obstruction and select appropriate surgical techniques for better postoperative outcomes.
